# Comprehensive Evaluation of Raw Eating Quality in 81 Sweet Potato (*Ipomoea batatas* (L.) Lam) Varieties

**DOI:** 10.3390/foods12020261

**Published:** 2023-01-06

**Authors:** Ximing Xu, Shiyu Wu, Kuangji Chen, Heyao Zhang, Shuke Zhou, Zunfu Lv, Yuantao Chen, Peng Cui, Zhongqiu Cui, Guoquan Lu

**Affiliations:** 1The Key Laboratory for Quality Improvement of Agricultural Products of Zhejiang Province, Institute of Root and Tuber Crops, College of Advanced Agricultural Sciences, Zhejiang A&F University, Hangzhou 311300, China; 2Yizheng Agricultural Technology Comprehensive Service Center, Yangzhou 211400, China; 3Key Laboratory of Crop Genetics and Breeding, Tianjin Crop Institute, Tianjin 300384, China

**Keywords:** raw eating quality, sweet potato, comprehensive evaluation, principal component analysis

## Abstract

The raw eating quality of sweet potato is complex. As consumers start paying more attention to the raw eating quality of tuberous roots in sweet potato, the evaluation of the raw eating quality of sweet potato is becoming an important issue. Therefore, we measured 16 quality indicators in 81 varieties of sweet potato. It was found that these 16 quality traits had different coefficients of variation (C.V.). Among them, the C.V. of fructose, glucose, and adhesiveness were the largest: 87.95%, 87.43% and 55.09%, respectively. The cluster analysis method was used to define six categories of the different tuberous roots of sweet potato. Group I, III, and IV had a stronger hardness and higher starch and cellulose content. Groups II, V, and VI were softer, with a high moisture and soluble sugar content. The principal component analysis method was used to comprehensively evaluate 16 quality indicators of 81 sweet potato varieties. It was found that Futian1, Taishu14, and Nanshu022 are good varieties in terms of raw eating quality. These varieties have low hardness, high adhesiveness in texture, high soluble sugar content, and low starch and cellulose. Future research should focus on improving the raw eating quality of sweet potato by reducing hardness, starch, and cellulose, while increasing adhesiveness, soluble sugar, and moisture content.

## 1. Introduction

As the world population continues to increase, more than 1.9 billion adults, 18 years and older, are now overweight, and 650 million are obese [1]. These medical conditions increase the risk of other diseases and health problems, such as heart disease, diabetes, high blood pressure, and certain cancers. Raw food diets are good methods for controlling body weight [2] since they are low in fat and contain protein and fiber, which help to keep the body in good shape. Raw food diets are also associated with favorable plasma β-carotene [3] and lower plasma, cholesterol, and triglyceride concentrations [4]. These diets often include the healthiest vegetarian cuisine and require careful planning and lifestyle adjustments, containing uncooked or raw foods such as fruits, vegetables, and legumes. Many healthy diets include raw foods, such as Mediterranean, Okinawa, DASH, etc., and raw root and tuber crops, such as yacon [5], yam bean, raw eating potato [6], and sweet potato [7], are high in biologically active substances, which may provide physiological benefits and nutritional functions. Sweet potato (*Ipomoea batatas* (L.) Lam) is a tuber crop and a nutritious and staple vegetable [8]. It has several special flavor and texture characteristics and is an excellent source of calories, carotene, vitamins, dietary fiber, etc. [9]. Many studies on the eating quality of sweet potato focus on roasted, boiled, and steamed sweet potatoes [10,11], while fewer studies address its quality as a raw vegetable. Therefore, an efficient and objective evaluation system is needed for raw sweet potatoes.

Raw sweet potatoes have less maltose sugar than those that have undergone heat processing because heating affects the starch’s transformation into maltose [12]. Cooked sweet potatoes are higher in sugar, as the heat turns starches into maltose for easier digestion, providing a sweeter flavor than raw sweet potatoes [13]. More sugar increases blood sugar after eating, causing concerns about blood pressure and diabetes [14]. Sweet-tasting tuberous roots of sweet potato are well-rounded nutritional powerhouses with a great deal of dietary fiber, minerals, vitamins, and antioxidants [15]. Therefore, eating raw sweet potato can effectively contribute to weight loss. However, selecting a raw sweet potato with a good flavor can be challenging.

Raw sweet potato is more similar to a fruit than a vegetable. Therefore, research on raw sweet potato can draw more lessons from the research on fruits and vegetables. The quality of fruit is often a very important factor for consumers. The measurement of raw crop (such as fruit and vegetable) attributes includes both internal and external characteristics [16,17]. Taste and texture are important factors in food evaluation since soluble sugars, organic acids and volatile compounds can affect taste. The soluble sugar concentration of fleshy fruit is a key determinant of its quality as this directly affects the sweetness of fruits and indirectly affects the properties of processed products [18]. Texture properties, including hardness, adhesiveness, cohesiveness, springiness, gumminess, and chewiness, are important factors. They are widely used for the evaluation of raw fruit [19,20]. These are the most direct sensory factors, and the texture characteristics of different sweet potato varieties differ. Therefore, the study of the taste and texture of tuberous roots of sweet potato varieties has important theoretical guiding significance for improving the raw quality of sweet potato.

A principal component analysis (PCA) is always used to comprehensively evaluate food quality, and this method helps researchers to determine high-quality raw sweet potato varieties. In this study, 16 quality indicators of 81 sweet potato varieties were tested. The quality of sweet potato (texture, sugar content, starch content, and moisture) was comprehensively evaluated to provide suggestions for improving the eating quality of raw sweet potato.

## 2. Materials and Methods

### 2.1. Plant Materials and Experiment Design

Eighty-one sweet potato varieties from the China Agriculture Research System (CARS) were used in this study, which was conducted in 2021 (Appendix A). Sweet potatoes were planted in May and harvested in October at CARS Sweet Potato Experiment Station, Hangzhou, Zhejiang, China. Essential information about these varieties is shown in Appendix A. The fertilizing treatment comprised 450 kg/hm^2^ NPK-15:15:15 complex fertilizer and 225 kg K_2_SO_4_. The trial was planted in three completely randomized blocks (about 12 m^2^), and 10 sweet potato seedlings were planted in each block of each variety. Tuberous roots were measured at Zhejiang A & F University after harvesting.

### 2.2. Textural Properties

The texture properties of tuberous roots of sweet potatoes were determined using a physical property analyzer (FTC company model TMS-PRO) with reference to the methods of Dong [21]. A 1 cm-thick disc was cut in the middle of the whole potato, and the TPA test was performed on the equatorial part of the disc using a physical property analyzer P/5 cylindrical probe (diameter 5 mm). Each variety was measured for 3 tubers, and each tuber was set up for 3 replicates. The sensory and mathematical definitions of the texture test indicators selected in this experiment are shown in Table 1.

### 2.3. Cell Wall Components Content

The hemicellulose, cellulose and lignin contents were measured using the wet chemistry methods of Van Soest [22]. Then, a 1.0 g air-dried sample was ground to pass through 40 mesh into a refluxing apparatus and weighed. The following were added in this order: 100 mL cold (room temperature) neutral-detergent solution, 2 mL decahydronaphthalene, and 0.5 g sodium sulfite with a scoop. The solution was heated to boiling in 5–10 min. The heat was reduced as boiling began in order to avoid foaming. The heat was adjusted to an even level and refluxed for 60 min, as timed from the onset of boiling. The beaker was swirled and the previously tared crucible was filled. A vacuum was used after the crucible was filled. A low vacuum was used at first, increasing only as more force was required. The sample was rinsed in the crucible with hot water at a minimum of 80–90 °C. The vacuum was removed, the mat broken up, and the crucible filled with hot water. Then, the liquid was filtered, and the washing procedure was repeated. The samples were then washed twice with acetone in the same manner and sucked dry. The crucible was dried at 100 °C for 8 h or overnight, before being cooled in an efficient desiccator and weighed. All of the reagents used in this study were of analytical grade. The relative error rate of the three replicate chemical measurements for each sample was maintained at less than 5%. Soluble pectin and protopectin were extracted and measured by Cao et al. [23].

### 2.4. Starch Content

Starch content was determined using an assay kit (Megazyme International Ltd., Bray, Co. Wicklow, Ireland) according to AACC Method 76-13.01 [24]. Powder samples (100 mg) were first washed with 80% ethanol to remove the sugars as described above. The remaining pellet was then treated with 2 mL of DMSO at 100 °C to account for the resistant starch. The samples were cooked with thermostable alpha amylase to partially hydrolyze and solubilize the starch. Subsequently, the samples were treated with amyloglucosidase for 30 min at 50 °C to hydrolyze the starch dextrins to glucose. The samples were then transferred to 100 mL volumetric flasks and filled to the brim with distilled water. An aliquot of this solution was centrifuged at 3000 rpm for 10 min, and the supernatant was mixed with a glucose determination reagent and incubated at 50 °C for 20 min. The absorbance of the solution at 510 nm was read on a T6-new century UV-visible spectrophotometer (Beijing Puxi Company, Beijing, China) against a reagent blank. Starch content was calculated based on the absorbance of the sample with reference to a glucose standard. Starch content was shown on a dry basis in this study.

### 2.5. Soluble Sugar Components

Soluble sugar components were determined by the method of Grabowski et al. [25]. A 0.2000 g sample was taken and washed twice with 8 mL ultrapure water in a centrifuge tube. After standing at room temperature for 2 h, the mixture was evenly vortexed and centrifuged at 4000 r/min for 10 min. Then, 1 mL supernatant was taken, added to 3 mL acetonitrile, and filtered through a 0.22 μm microporous membrane for chromatographic analysis. The detection equipment used was an Agilent 1260 Infintiy II high-performance liquid chromatograph and an AllChrom ELSD 6000 evaporative light scattering detector. The PrevailTM Carb ES Coumn-W 250 mm × 4.6 mm column was used as the stationary phase and acetonitrile–water (*v*/*v* = 65:35) as the mobile phase, with a flow rate of 0.8 mL·min^−1^, column temperature of 30 °C, drift tube temperature of 95 °C, nitrogen flow rate of 2.4 mL·min^−1^, and injection volume of 4.0 μL.

### 2.6. Moisture Content

Moisture content was determined by the AACC international method 44-19.0 [26]. The sweet potato samples were sliced, and 10.00 g of sweet potato was weighed in a covered dish previously dried at 135 °C, cooled in a desiccator, and weighed again soon after reaching room temperature. The samples on the dish in the baking oven were dried at 135 °C for at least 2 h. Then, the samples were cooled in a desiccator and weighed soon after attaining room temperature. The moisture content of tuberous roots was calculated.

### 2.7. Statistical Analysis

The SPSS 23.0 software (IBM, Chicago, IL, USA) was used for the analysis of variance (ANOVA), and Origin 2021 (OriginLab, Northampton, MA, USA) was used for the mapping and correlation analysis. All data in the experiment were analyzed in triplicate. For the principal component analysis (PCA), the quality attributes of different sweet potato varieties were normalized, the characteristic value and contribution rate were determined, and the raw eating quality score of sweet potatoes was calculated.

## 3. Results

### 3.1. Texture Properties

Texture properties are a series of comprehensive concepts that can accurately determine the quality of sweet potato [27]. The hardness of tuberous roots is an important aspect of its texture properties, and a sweet potato with a high raw eating quality tends to have a low hardness. In terms of texture properties, the hardness of tuberous roots varies from 72.82 N to 142.26 N, with an average value of 105.86 N (Table 2). The hardness of Ning B73-8 (142.26 N), Chuanshu231 (137.79 N), and Xuzishu8 (135.74 N) was greater than other varieties, and in this study, Futian1 (72.82 N), Jishu29 (76.30 N), and Eshu17 (81.87 N) were softer than other varieties (Appendix A.). The adhesiveness of the tuberous roots varied from 0.91 (Xushu18) to 12.27 mJ (Sushu25), with an average value of 5.08 mJ (Table 2). The cohesiveness of the tuberous roots varied from 0.17 (Ningzishu6) to 0.29 (Xushu18), with an average value of 5.08 mJ (Table 2). The springiness of the tuberous roots varied from 3.92 (WanA4921) to 7.78 mm (Luozishu6), with an average value of 5.11 mm. The gumminess of the tuberous roots varied from 15.80 (Futian1) to 29.68 N (Xuzishu8), with an average value of 21.82 N. The chewiness of the tuberous roots varied from 73.96 (WanA4921) to 172.89 mJ (Xushu18), with an average value of 111.95 mJ. The coefficients of variation (C.V.) of these texture properties were 13.38%, 55.09%, 10.02% 9.76%, 14.48%, and 17.73%. The above indicators show significant differences among the varieties in this study (*p* < 0.01) (Table 2).

### 3.2. Soluble Sugar

Soluble sugar is the main type of sugar in the tuberous roots of sweet potato and includes fructose, sucrose, and glucose. The soluble sugar level determines the taste quality of raw sweet potato, and different varieties have different soluble sugar fractions. As Table 2 shows, the fructose content varied from 3.42 (Zheshu75) to 113.50 mg·g^−1^ (Taishu14), with an average value of 24.64 mg·g^−1^. The sucrose content of tuberous roots varied from 22.50 (Qining19) to 146.41 mg·g^−1^ (Zheshu21), with an average value of 86.42 mg·g^−1^. The glucose content of tuberous roots varied from 3.81 (Zheshu75) to 131.48 mg·g^−1^ (Taishu14), with an average value of 27.81 mg·g^−1^. The C.V. values of soluble sugar were 87.43%, 27.96%, and 87.95%. The above indicators show greater differences among varieties in this study (Appendix A). The average value of sucrose content was 86.42 mg·g^−1^. This accounts for the largest proportion of soluble sugar content and is consistent with the findings of previous studies [28]. The fructose and glucose contents of Taishu 14, Futian 1, and Zhanshu 407 were higher than other varieties in this study. The fructose and glucose content of C.V. was more prominent than sucrose content. Fructose and glucose content can better represent the raw eating quality of varieties (Table 2 and Appendix A).

### 3.3. Cell Wall Components

The plant cell wall is composed of polysaccharide polymers and other substances. Its polysaccharide polymers mainly include lignin, pectin, cellulose, and hemicellulose, which provide mechanical support and protection for cells and maintain their expansion [29]. Lignin is a natural phenolic polymer with a high molecular weight and a complex composition and structure [30], and this substance may affect the hardness and chewiness of the tuberous root. The lignin content of tuberous roots varies from 0.38 (Longshu9) to 3.53 mg·g^−1^ (Miannanshu10) in this study, and the C.V. value is 28.74%. Soluble pectin had a stronger correlation with the adhesiveness of tuberous roots [19]; its content in tuberous roots varied from 0.84 (Jiyuan1) to 2.56 mg·g^−1^ (Eshu19), and its C.V. was 25.00%. Hemicelluloses are polysaccharides in plant cell walls that have beta-(1→4)-linked backbones with an equatorial configuration. The most important biological function of hemicellulose is to strengthen the cell wall through the interaction with cellulose and lignin [31]. Hemicellulose content varies from 158.43 (Qining26) to 304.40 mg·g^−1^ (Futian1), and C.V. is 15.10%. Protopectin content varies from 10.38 (Longshu24) to 18.31 mg·g^−1^ (Wansu4723), and C.V. is 13.20%. Cellulose is a polysaccharide composed of linear glucan chains that are linked together by β-1,4-glycosidic bonds with cellobiose residues as the repeating units at different degrees of polymerization [32]. Cellulose content varied from 65.77 (Wansu4723) to 78.68 mg·g^−1^ (Miannanshu10), and C.V. is 3.99%, and in this study, the above indicators showed significant differences among varieties (*p* < 0.01). The order of C.V. is lignin > soluble pectin > hemicellulose > protopectin > cellulose (Table 2 and Appendix A).

### 3.4. Moisture and Starch Content

Moisture content is an important aspect of the tuberous roots of sweet potatoes and varies from 60.11 (NingB73-8) to 86.31% (Futian1). In this study, the C.V. of moisture content was 7.26%. Starch content makes up a large proportion of the dry tuberous root of sweet potato, varying from 41.23 (Futian1) to 86.56% (Sushu29). The C.V. of starch content was 11.86%. The above indicators show significant differences among the varieties used in this study (*p* < 0.01) (Table 2 and Appendix A).

### 3.5. Correlation Analysis

As shown in Figure 1, 16 quality indicators were correlated to some extent. There was no significant correlation between lignin content, soluble pectin content, and cellulose content regarding texture indicators. Hardness had a negative correlation with cohesiveness, fructose, sucrose, glucose, hemicellulose and moisture content (R = −0.259, *p* < 0.05; R = −0.512, R = −0.405, R = −0.501, R = −0.470, R = −0.760, *p* < 0.01, respectively) and a stronger positive correlation with gumminess, chewiness, and starch content (R = 0.752, R = 0.509, R = 0.537, respectively; *p* < 0.01). This indicates that the starch content accumulated in tuberous roots may increase in hardness, while fructose, sucrose, glucose, hemicellulose, and moisture content decrease.

Adhesiveness has a strong positive correlation with cohesiveness, springiness, and fructose, glucose, soluble pectin, and hemicellulose content (R = 0.286, R = 0.320, R = 0.328, R = 0.329, R = 0.326, and R = 0.387, respectively; *p* < 0.01) and a negative correlation with starch content (R = −0.333; *p* < 0.01). This indicates that a high adhesiveness accompanies a high soluble content, especially regarding glucose and fructose content. Cohesiveness has a strong positive correlation with springiness, chewiness, fructose, glucose, hemicellulose, moisture, and soluble pectin content (R = 0.268, R = 0.429, R = 0.328, R = 0.515, R = 0.350, R = 0.342, R = 0.343, and R = 0.357, *p* < 0.01; R = 0.265, *p* < 0.05, respectively). There was a negative correlation between cohesiveness and starch content (R = −0.348, *p* < 0.01), and springiness had a strong negative correlation with starch content (R = −0.249, *p* < 0.05). There was a stronger positive correlation between gumminess and chewiness (R = 0.818, *p* < 0.01) and a strong negative correlation between gumminess and fructose, sucrose, glucose, and moisture content (R = −0.247, R = −0.257, R = −0.242, *p* < 0.05; R = −0.471, *p* < 0.05, respectively). Chewiness had a strong negative correlation with moisture content (R = −0.329, *p* < 0.01), indicating high moisture content; a lower starch content in tuberous roots makes it easier to chew when raw.

### 3.6. Cluster Analysis

A cluster analysis is used to classify objects with the same attributes into a single category in order to better predict the attributes of different categories. Firstly, the quality of 81 tuberous roots of sweet potato was standardized to remove the influence of units, and then a cluster analysis was performed based on inter-group connections and the Pearson correlation. A Euclidean distance of 20 was used to divide the roots into six groups (Figure 2).

Group I contained 24 types of sweet potato, including Benimasari, Qining 16, Longshu14, etc. These varieties have a high hardness and starch content and a low sucrose, hemicellulose, and moisture content (Figure 3, Figure 4, Figure 5 and Figure 6).

Only four varieties, Nongdahong, Xushu18, Zheshu81, and Qining37, were categorized into Group II. The main features of these varieties were lower adhesiveness and soluble pectin than other varieties (Figure 3 and Figure 5), while cohesiveness, springiness, gumminess, and chewiness all had high values (Figure 3). Fructose and glucose content was higher than in other groups, except Group V (Figure 4).

Mianzishu15, WanA491, Zheshu27, Wansu546, Xushu44, Pushu32 (Xiguahong), and Sushu33 were categorized into Group III. Their main features are a low springiness in texture, high sucrose, low fructose and glucose content in soluble sugar, high soluble pectin and low cellulose content in cell wall components, and high starch content (Figure 3).

Group IV contains 18 varieties, including Qining19, Xinxiang, Xushu37, etc. Their main features are low cohesiveness, gumminess, and chewiness in terms of texture, low sucrose in soluble sugar, high lignin and cellulose content and low protopectin content in cell wall components, low moisture content, and high starch content (Figure 3).

Group V contains 17 varieties, including Futian1, Longshu5, Jishu33, etc. Their main features are low hardness, gumminess, and chewiness in terms of texture, low sucrose and high fructose and glucose content in soluble sugar, low lignin and high protopectin content in cell wall components, high moisture content, and low starch content (Figure 3).

Group VI contains 11 varieties, including Qining17, Sushu9, Eshu19, etc. Their main features are high adhesiveness in terms of texture, high sucrose content in soluble sugar, low cellulose and high protopectin and hemicellulose content in cell wall components, high moisture content, and low starch content (Figure 3).

The six groups have distinct characteristics. It is worth noting that Groups I, III, and IV have a greater hardness and higher starch and cellulose contents. Group II, V, and VI are soft, juicy, and have high soluble sugar contents.

### 3.7. Principal Component Analysis (PCA)

PCA is the most commonly used dimension reduction method. Using this method, the correlated variables are converted into uncorrelated principal components (PCs), which allows the maximum variance between features to be recorded [33]. In this study, the Kaiser–Meyer–Olkin (KMO) value was 0.653, and the significance level was 0.000 (Appendix A), indicating that PCA could be performed. Five components accounted for 82.342% of the data difference, and their contribution rates were 33.692%, 19.125%, 11.920%, 11.317%, and 6.288% (Table 3 and Appendix A). The highest characteristic eigenvalue of PC1 was 5.391. PC1 was highly positively correlated with moisture and hemicellulose content, and negatively correlated with starch content and hardness, referred to as the moisture factor. The characteristic value of PC2 was 3.060, and the gumminess, chewiness, and cellulose characteristics corresponded to the highest eigenvector, known as the chewiness factor. The characteristic value of PC3 was 1.907, and adhesiveness and lignin and protopectin content corresponded to the highest characteristic vector, which mainly reflected what we referred to as the stickiness factor. The characteristic value of PC4 was 1.811. PC4 was highly positively correlated with fructose and glucose content and negatively correlated with sucrose and soluble pectin content, and mainly contained sugar, and thus was named the sugar factor. The characteristic value of PC5 was 1.006, and the cohesiveness and springiness corresponded to the highest eigenvector, known as the cohesiveness factor. Then, the five main components were summarized as moisture, chewiness, stickiness, sugar, and cohesiveness.

### 3.8. Comprehensive Evaluation of Raw Eating Quality

We analyzed data and found that high hardness and cellulose content were not good for raw eating. Therefore, PC1 and PC4 have positive values, and PC2, PC3, and PC5 have negative values. We calculated the principal component model by taking the ratio of the five PCs to the eigenvalues. The sum of the eigenvalues of all PCs was as follows:Y_n_ = PC_n1_ × X_1_ + PC_n2_ × X_2_ + PC_n3_ × X_3_… + PC_n16_ × X_16_(1)
F = Y_1_ × 0.409 − Y_2_ × 0.232 − Y_3_ × 0.145 + Y_4_ × 0.138 − Y_5_ × 0.076(2)

Among these values, PC_n_ represents the feature vector of the corresponding matrix; X_n_ represents the standardized index of raw quality; Y_n_ represents the score of each main component in the tuberous roots; and F represents the comprehensive score of raw eating quality.

From the perspective of comprehensive scores, the score types with a comprehensive score greater than 1 were selected, and a total of three varieties were found, with the highest ranking being Futian1 (Appendix A). The second- and third-ranked varieties were Taishu14 and Nanshu022. These varieties of tuberous roots are characterized by a lower hardness, cellulose content, and starch content, and higher fructose, glucose, and soluble pectin content.

## 4. Discussion

The raw eating quality of tubers and root crops requires different evaluations. The heating process of tuberous roots affects the starch gradually gelatinizing, leading to the leaching of maltose [13]. Then, the sugar content of sweet potato increases and increases the risk associated with sweet potato consumption for diabetics. Small-starch-granule varieties of tubers and root crops are more suitable for raw eating, such as sweet potato, taro, and potato [6,7,34]. The evaluations of roasted, steamed, and boiled sweet potatoes are more similar to evaluations of vegetables or fruits, such as lettuce, apple, etc. [35,36], and can easily be affected by different genotypes [37]. Hardness is one of the most important texture properties and is most easily affected by accumulated starch, and is positively correlated with the starch content at the maturity stage. Previous research on the texture of tuberous roots found that the generation of starch is related to the carbohydrate metabolism of sweet potato, and the increase in starch accumulation is related to the hardness of tuberous roots [38]. Interestingly, previous research found that different cultivation periods may affect the texture properties of tuberous roots [21]. This indicates that carbohydrate metabolism is essential for changing the textural properties of tuberous roots.

The cellulose fraction was affected by soluble pectin content and texture properties [39]. Meanwhile, the change in cell wall structure and composition is the main reason for the change in fruit texture. In this study, we found similar trends: texture properties had a stronger correlation with cell wall composition, and soluble pectin and hemicellulose content had a strong positive correlation with adhesiveness and cohesiveness. Moreover, there was a very significant negative correlation between moisture content and the hardness of tuberous roots (*p* < 0.01). Starch content had a strong negative correlation with adhesiveness and cohesiveness.

Soluble pectin, hemicellulose, moisture, and starch could be essential factors that affect the textural properties of different sweet potato varieties. After a cluster analysis, we divided 81 varieties into six different groups. Groups I, III, and IV contain firmer sweet potatoes with a higher starch content than other groups, making them appropriate for cooking or food processing. Groups II, V, and VI contain softer sweet potatoes with a higher soluble sugar content than other groups, making them appropriate for raw eating or cooking. According to the market demand, juicy, sweet, and palatable fruit have high-quality taste attributes [40,41,42]. We believe that soft sweet potatoes with high moisture, fructose, and glucose content, as well as a low cellulose content, are suitable for eating raw. We used PCA to extract five factors with characteristic values >1 from 16 quality indicators. The cumulative contribution rate of variance for the five common factors was 82.342%. The final model was constructed to screen sweet potato varieties with good taste that can be eaten raw. This study provides a theoretical basis for the directional selection of sweet potato varieties that can be eaten raw. A high score indicates that these varieties are suitable for eating raw. In this study, we found that Futian1, Taishu14, and Nanshu022 have higher raw eating comprehensive scores than other varieties.

## 5. Conclusions

In sweet potato varieties, several factors can have very complex effects on raw eating quality. It is necessary to consider the texture, sweetness, and juiciness of sweet potato for the evaluation of raw eating quality. We evaluated changes in sixteen qualities of 81 varieties under the same cultivated environment. Among these qualities, the coefficient of variation, glucose content, fructose content, and adhesiveness were the largest: 87.95%, 87.43%, and 55.09%, respectively. This indicated that glucose content, fructose content, and adhesiveness are the most susceptible characteristics of sweet potato varieties. A cluster analysis divided sweet potatoes into six categories with different qualities. Each category had its own unique quality attributes. The PCA revealed the raw eating quality comprehensive evaluation scores of different sweet potato varieties. The eating quality of Futian1, Taishu14, and Nanshu022 was excellent. Future research on raw sweet potato should focus on reducing hardness and starch and cellulose content, and increasing fructose, glucose, and moisture content in order to improve the raw eating quality of tuberous roots in sweet potato.

## Figures and Tables

**Figure 1 foods-12-00261-f001:**
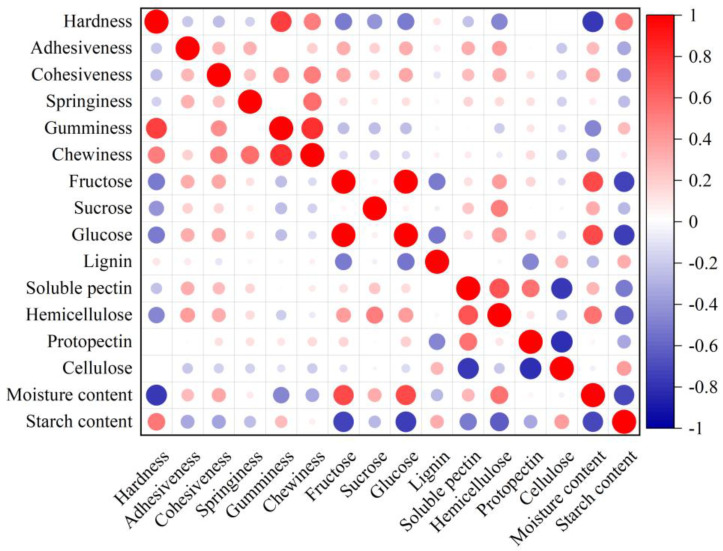
Correlation analysis of quality traits of sweet potato tuberous roots (*n* = 81).

**Figure 2 foods-12-00261-f002:**
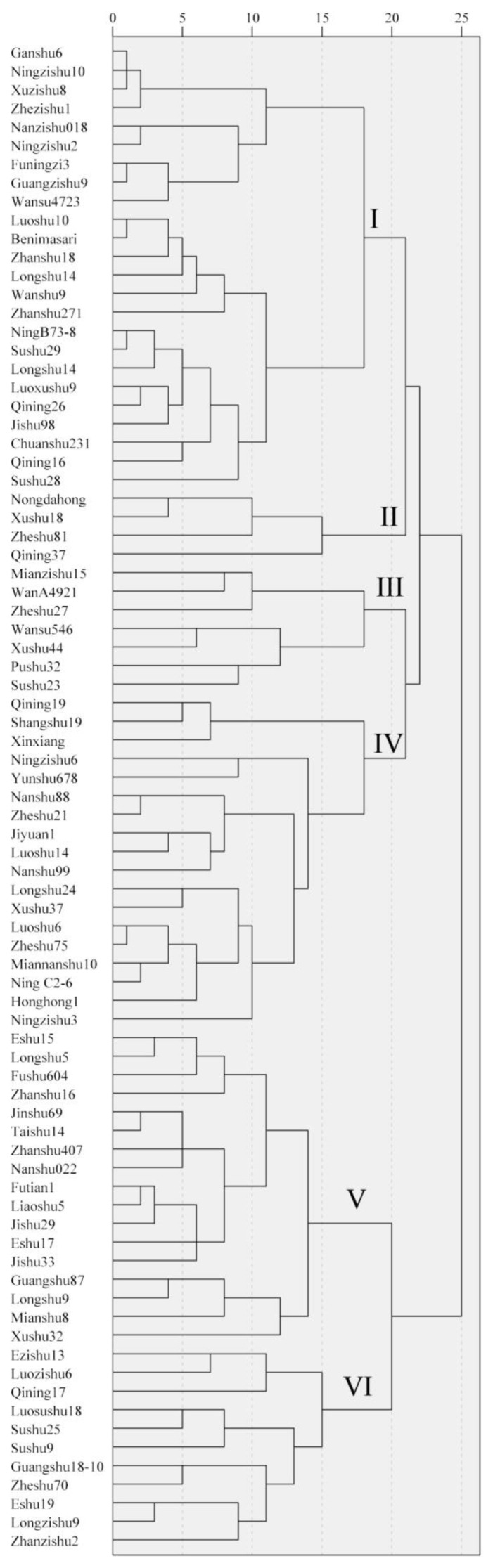
Cluster analysis of quality traits in the tuberous roots of 81 sweet potato varieties. Roman numerals I to VI indicate six groups.

**Figure 3 foods-12-00261-f003:**
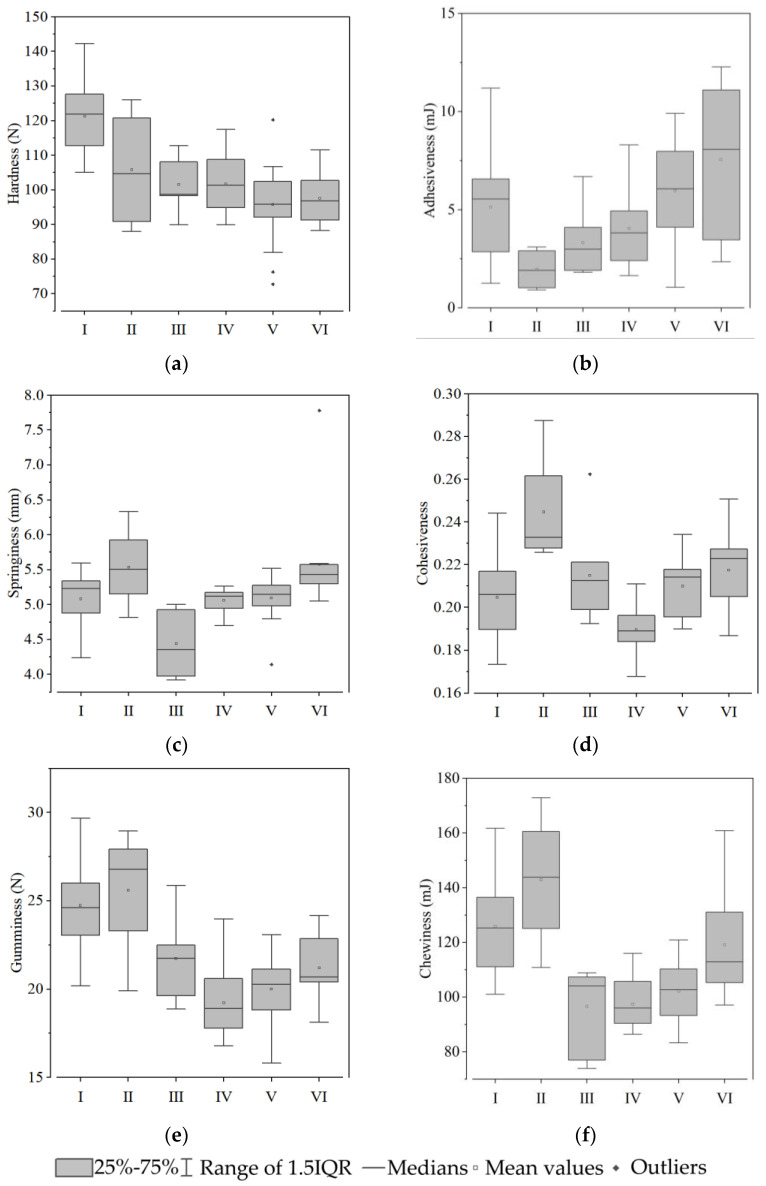
Textural properties of the tuberous roots of six sweet potato characteristics: (**a**) hardness; (**b**) adhesiveness; (**c**) springiness; (**d**) cohesiveness; (**e**) gumminess; and (**f**) chewiness. Hollow squares indicate mean values. Rhombuses indicate outliers.

**Figure 4 foods-12-00261-f004:**
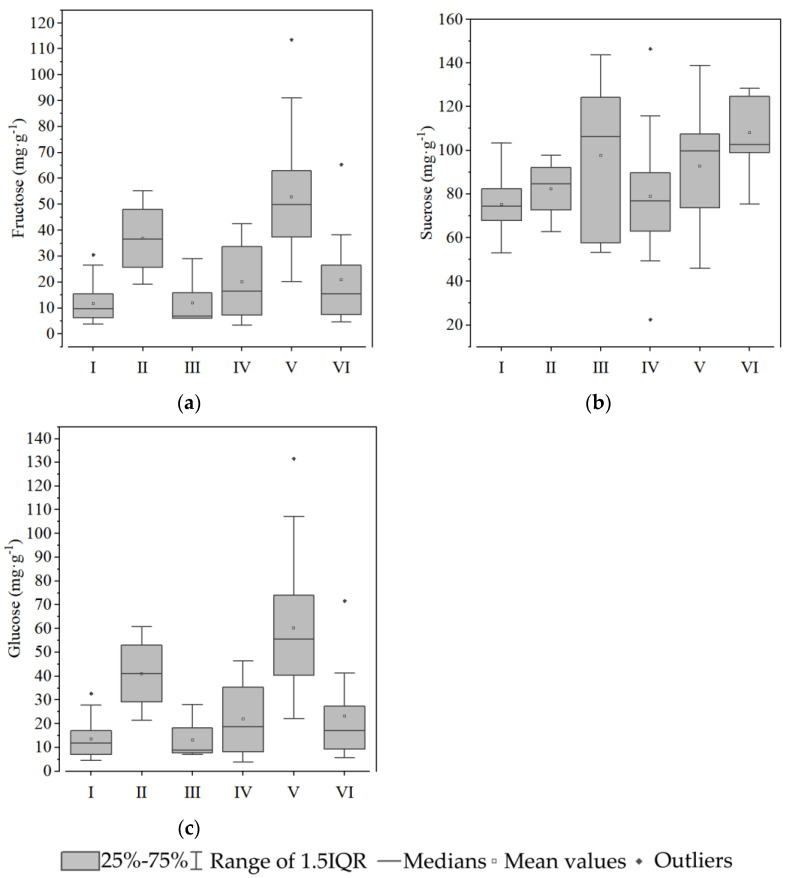
Soluble sugar components of the tuberous roots of six groups of sweet potato: (**a**) fructose; (**b**) sucrose; and (**c**) glucose. White squares indicate mean values. Rhombuses indicate outliers.

**Figure 5 foods-12-00261-f005:**
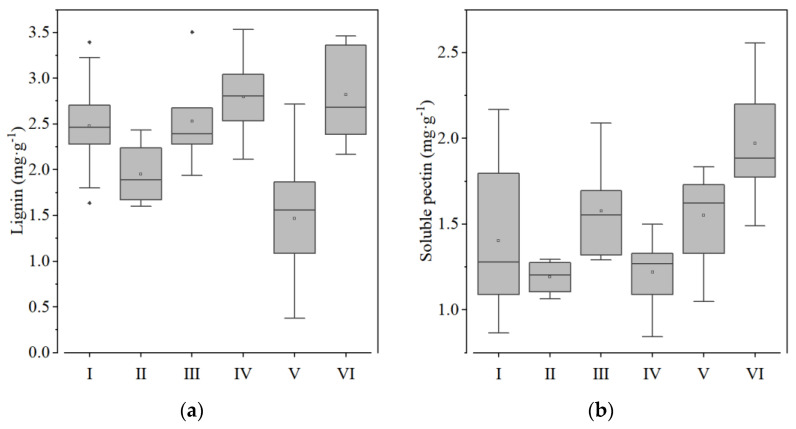
Cell wall components of the tuberous roots of six groups of sweet potato: (**a**) lignin; (**b**) soluble pectin; (**c**) hemicellulose; (**d**) protopectin; and (**e**) cellulose. White squares indicate mean values. Rhombuses indicate outliers.

**Figure 6 foods-12-00261-f006:**
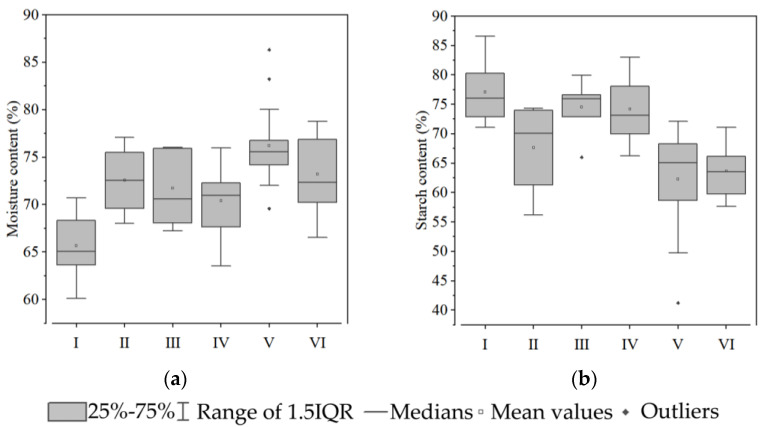
Moisture and starch content of the tuberous roots of six groups of sweet potato: (**a**) moisture content; and (**b**) starch content. White squares indicate mean values. Rhombuses indicate outliers.

**Table 1 foods-12-00261-t001:** Definition of the parameters measured by texture analyzer.

Parameters	Definition	Unit
Hardness	Maximum strength peak for the first extrusion cycle	N
Adhesiveness	The area of the curve in the negative direction of the coordinate axis between two extrusion cycles	mJ
Springiness	The ratio of the positive peak area of the second extrusion cycle to the positive peak area of the first extrusion cycle	mm
Cohesiveness	The ratio of the height of the second compression to that of the first compression	Ratio
Gumminess	Hardness × Cohesiveness	N
Chewiness	Hardness × Cohesiveness × Springiness	mJ

**Table 2 foods-12-00261-t002:** Descriptive statistics of sweet potato tuberous root quality changes.

Parameters	Range	Min	Max	Mean	S.D.	C.V.	Variety
Hardness (N)	69.44	72.82	142.26	105.86	14.16	13.38%	**
Adhesiveness (mJ)	11.36	0.91	12.27	5.08	2.80	55.09%	**
Cohesiveness	0.12	0.17	0.29	0.21	0.02	10.02%	**
Springiness (mm)	3.86	3.92	7.78	5.11	0.50	9.76%	**
Gumminess (N)	13.88	15.80	29.68	21.82	3.16	14.48%	**
Chewiness (mJ)	98.93	73.96	172.89	111.95	19.85	17.73%	**
Fructose (mg⋅g^−1^)	110.08	3.42	113.50	24.64	21.54	87.43%	**
Sucrose (mg⋅g^−1^)	123.91	22.50	146.41	86.42	24.16	27.96%	**
Glucose (mg⋅g^−1^)	127.66	3.81	131.48	27.81	24.46	87.95%	**
Lignin (mg⋅g^−1^)	3.15	0.38	3.53	2.36	0.68	28.74%	**
Soluble pectin (mg⋅g^−1^)	1.71	0.84	2.56	1.47	0.37	25.00%	**
Hemicellulose (mg⋅g^−1^)	145.97	158.43	304.40	223.63	33.77	15.10%	**
Protopectin (mg⋅g^−1^)	7.93	10.38	18.31	13.40	1.77	13.20%	**
Cellulose (mg⋅g^−1^)	12.92	65.77	78.68	73.11	2.92	3.99%	**
Moisture content (%)	26.20	60.11	86.31	70.82	5.14	7.26%	**
Starch content (%)	45.33	41.23	86.56	70.82	8.40	11.86%	**

Note: Range, difference between maximum and minimum; Min, minimum; Max, maximum; Mean, mean of all samples; SD, standard deviation of all samples; CV, coefficient of variation for all samples; ** significant differences at the 0.01 threshold.

**Table 3 foods-12-00261-t003:** Eigenvectors of corresponding matrices for tuberous root quality traits of sweet potato.

Indicators	PC1	PC2	PC3	PC4	PC5
Hardness (N)	−0.316	0.271	−0.108	0.093	0.131
Adhesiveness (mJ)	0.193	0.136	0.319	−0.034	−0.206
Cohesiveness	0.186	0.280	0.290	0.205	0.362
Springiness (mm)	0.107	0.235	0.283	0.035	−0.686
Gumminess (N)	−0.167	0.443	0.097	0.208	0.373
Chewiness (mJ)	−0.074	0.493	0.247	0.196	−0.087
Fructose (mg·g^−1^)	0.349	−0.055	−0.055	0.383	−0.013
Sucrose (mg·g^−1^)	0.172	−0.084	0.208	−0.309	0.305
Glucose(mg·g^−1^)	0.352	−0.044	−0.072	0.373	−0.014
Lignin (mg·g^−1^)	−0.166	−0.042	0.442	−0.324	−0.090
Soluble pectin (mg·g^−1^)	0.240	0.263	−0.078	−0.447	0.066
Hemicellulose (mg·g^−1^)	0.307	0.034	0.220	−0.280	0.249
Protopectin (mg·g^−1^)	0.149	0.310	−0.470	−0.135	−0.076
Cellulose (mg·g^−1^)	−0.167	−0.353	0.351	0.277	0.103
Moisture content (%)	0.365	−0.168	0.075	0.073	0.087
Starch content (%)	-0.382	-0.043	0.028	-0.029	0.025
Eigenvalue	5.391	3.060	1.907	1.811	1.006
Percentage of variance (%)	33.692	19.125	11.920	11.317	6.288
Cumulative (%)	33.692	52.817	64.737	76.055	82.342
Weight coefficient	0.409	0.232	0.145	0.138	0.076

## Data Availability

The data are available from the corresponding author. The data are not publicly available due to provisions of China Agriculture Research System.

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
