# Peer review of "Comprehensive Evaluation of Raw Eating Quality in 81 Sweet Potato (Ipomoea batatas (L.) Lam) Varieties"

_foods, 2023, doi:10.3390/foods12020261_

Round 1

Reviewer 1 Report

General opinion on the paper:

Raw-eating type (Fruit type) sweet potato comprehensive Evaluation of 16 Qualities of 81 varieties Based on Principal Component Analysis

The paper is well written, it has scientific soundness and the authors have expressed good knowledge in data processing and presenting their results. 

Some changes should be done so the detailed opinion is to create the following changes upon the reviewer's request: 

Raw food diets are good methods for weight control [2]. State which diets are good methods for weight control, e.i. Okinawa, Mediterranean diet ….

In line 38 explain what did you mean by “Raw food diet offers the ultimate vegetarian fare.

In lines, 39-40 give some figures if you claim that the market rises, a % or Number, and some literature/statistics with figures for China and Korea

“Therefore, the demand for raw-eating root and tuber crops were increasing sharply in the market, such as yacon, yam bean, and raw-eating potato (fruit-potato) in 40 China market[5, 6], raw-eating sweet potato (fruit-sweet potato) in the Korean market[7].

Line 51 provide evidence for the claim “Cooking promotes the starches to transform into maltose in the tuberous root of sweet potatoes”. e.i. some literature or findings.

In line 54 very vague hypothesis: “More soluble sugar would increase blood sugar after eating, which may make people worry about blood pressure and diabetes”

If You claim this show evidence that people think so – a case study or wide population study

If not erase this part

In methodology:

Line 83 They were carried out in May and…  (change the word carried out to planted)

I suppose the potato was a planted in may and then harvested in October

Line 100

Soluble pectin content was determined by AOAC…

State some details of the analysis, similar to the description in 2.4. moisture content

The results and conclusions are very well explained so no changes are required. 

Best regards, 

the reviewer 

Author Response

On behalf of my co-authors, we are very grateful to you for giving us an opportunity to revise our manuscript. We appreciate you very much for your positive and constructive comments and suggestions on our manuscript entitled “Raw-eating type (Fruit type) sweetpotato comprehensive Evaluation of 16 Qualities of 81 varieties Based on Principal Component Analysis” (ID: foods-2043190). We have studied the reviewers’ comments carefully and tried our best to revise our manuscript according to the comments. The following are the responses and revisions I have made in response to the reviewers' questions and suggestions on an item-by-item basis. Thanks again for the hard work of the editor and reviewer!

Point 1: Raw food diets are good methods for weight control [2]. State which diets are good methods for weight control, e.i. Okinawa, Mediterranean diet …. In line 38 explain what did you mean by “Raw food diet offers the ultimate vegetarian fare. In lines, 39-40 give some figures if you claim that the market rises, a % or Number, and some literature/statistics with figures for China and Korea. “Therefore, the demand for raw-eating root and tuber crops were increasing sharply in the market, such as yacon, yam bean, and raw-eating potato (fruit-potato) in 40 China market[5, 6], raw-eating sweet potato (fruit-sweet potato) in the Korean market[7].

Response 1:

Thanks for your suggestion. We followed this suggestion to revise. The sentence we revised:

Raw food diet (Raw foodism) is a way of eating that includes only uncooked or raw foods, as fruits, vegetables, and legumes, the raw food diet takes careful planning and lifestyle adjustments. Many healthy diet styles can find raw food diet, such as Medi-terranean, Okinawa, DASH, etc. Raw-eating root and tuber crop had more biologically active components that may provide physiological benefits and nutritional functions, such as yacon[5], yam bean, raw-eating potato[6] and sweetpotato[7].

Point 2:

Line 51 provide evidence for the claim “Cooking promotes the starches to transform into maltose in the tuberous root of sweet potatoes”. e.i. some literature or findings.

Response 2:Thanks for your suggestion. We followed this suggestion to revise. The sentence we revised: Raw-eating type sweetpotato was can be used widely in raw food diet. Raw sweet potatoes aren’t affected by heat, so and they’ll have less sugar than cooked. Cooking promotes the starches to transform into maltose in the tuberous root of sweetpotatoes[12].

And we add a new reference: Wei, S. Y.; Lu, G. Q.; Cao, H. P., Effects of cooking methods on starch and sugar composition of sweetpotato storage roots. Plos One 2017, 12, (8).

Point 3: In line 54 very vague hypothesis: “More soluble sugar would increase blood sugar after eating, which may make people worry about blood pressure and diabetes” If You claim this show evidence that people think so – a case study or wide population study.If not erase this part.In methodology:

Response 3:Thanks for your suggestion. We followed this suggestion to revise. The sentence we revised: More sugar would increase blood sugar after eating, which may make people worry about the blood pressure and diabetes [14].

We add a reference: Prada, M.; Saraiva, M.; Garrido, M. V.; Serio, A.; Teixeira, A.; Lopes, D.; Silva, D. A.; Rodrigues, D. L., Perceived Associations between Excessive Sugar Intake and Health Conditions. Nutrients 2022, 14, (3).

Point 4: Line 83 They were carried out in May and…  (change the word carried out to planted) I suppose the potato was a planted in may and then harvested in October

Response 4: Thanks for your suggestion. We followed this suggestion to revise. The sentence we revised:

They were carried outplanted in May and harvested in October 2021 at CARS Sweet-potato Experiment Station, Hangzhou, Zhejiang, China.

Point 5: Line 100. Soluble pectin content was determined by AOAC State some details of the analysis, similar to the description in 2.4. moisture content. The results and conclusions are very well explained so no changes are required.

Response 5: Thanks for your suggestion. We followed this suggestion to revise. The sentence we revised: They were planted in May and harvested in October 2021 at CARS Sweet-potato Experiment Station, Hangzhou, Zhejiang, China.

In general, we think your opinion is constructive. Thanks again for your suggestions.

Reviewer 2 Report

A summary and broad comments: The results can be of interest to the readers. The manuscript requires some minor adjustments (see the specific comments below)

Specific comments:

Materials and Methods

Lines 91-95: “Hardness..” The sentence is too long and incomprehensible. Please correct it.

Lines 100-108: Include all international methods used in the references.

Lines 107-112: Moisture content – It is the standard procedure, and its description is not necessary.

Line 113: Statistical analysis – Please include the number of repetitions of the analyses performed (used for the mean calculation)

Results

Line 135: “… average value of 19.85” – I should be 111.95. Please correct it.

Line 142 and 154: These two sentences repeat the same information regarding the proportion of fructose and glucose among sweet potato sugars. Please correct it.

Line 196 and 201: “stronger positive correlations” – stronger than what?. I suppose it should be “strong” instead of “stronger”. Please define why you use the word strong when the R-value is between 0.2-0.4

Fig 1 is too small. Difficult to understand the information it shows. Please correct it.

Line 242 „low springiness” (Group III) and line 246 “low cohesiveness” (Group IV): It seems to me that it should be “low cohesiveness” for Group III and „low springiness” for Group IV. Please check and correct if needed.

Line 259: “sweet” – Your study did not evaluate the sweetness of sweet potato varieties. I suggest changing to „high soluble sugar contents”.

Author Response

On behalf of my co-authors, we are very grateful to you for giving us an opportunity to revise our manuscript. We appreciate you very much for your positive and constructive comments and suggestions on our manuscript entitled “Raw-eating type (Fruit type) sweetpotato comprehensive Evaluation of 16 Qualities of 81 varieties Based on Principal Component Analysis” (ID: foods-2043190).We have studied reviewers’ comments carefully and tried our best to revise our manuscript according to the comments. The following are the responses and revisions I have made in response to the reviewers' questions and suggestions on an item-by-item basis. Thanks again to the hard work of the editor and reviewer!

Point 1:

Lines 91-95: “Hardness..” The sentence is too long and incomprehensible. Please correct it.

Response 1:

Thanks for your valuable comments. We followed this suggestion to revise. We add a table to explain this.

Table 1 Definition of the parameters measured by texture analyzer

Parameters

Definition

Unit

Hardness

Maximum strength peak for the first extrusion cycle

N

Adhesiveness

The area of the curve in the negative direction of the coordinate axis between two extrusion cycles

mj

Springiness

The ratio of the positive peak area of the second extrusion cycle to the positive peak area of the first extrusion cycle

mm

Cohesiveness

The ratio of the height of the second compression to that of the first compression

Ratio

Gumminess

Hardness × Cohesiveness

N

Chewiness

Hardness × Cohesiveness × Springiness

mj

Point 2:

Lines 100-108: Include all international methods used in the references.

Response 2: Thanks for your valuable comments.

Point 3:

Moisture content – It is the standard procedure, and its description is not necessary.

Response 3: Thanks for your valuable comments. We politely disaggreed due to the rules of journal.

Point 4: Line 113: Statistical analysis – Please include the number of repetitions of the analyses performed (used for the mean calculation)

Response 4: Thanks for your suggestion. We followed this suggestion to revise. This part we revised: The SPSS 23.0 software (Chicago, IL, USA) and Origin 2021 (Northampton, MA, USA) were used for the analysis of variance (ANOVA), mapping, and correlation analysis.  All the data in the experiment were analyzed triplicates. For principal component analysis (PCA), the quality properties of different sweetpotato varieties were normalized, the characteristic value and contribution rate were determined, and the raw-eating quality score of sweetpotato was formed.

Point 5: Line 135: “… average value of 19.85” – I should be 111.95. Please correct it.

Response 5: Oops, this is our negligence. Thanks for your comment. We have corrected it.

Point 6: Line 142 and 154: These two sentences repeat the same information regarding the proportion of fructose and glucose among sweet potato sugars. Please correct it.

Response 6: Oops, this is our negligence. Thanks for your comment. We have corrected it. The sentence of line 154 we have delelted.

Point 7: Line 196 and 201: “stronger positive correlations” – stronger than what?. I suppose it should be “strong” instead of “stronger”. Please define why you use the word strong when the R-value is between 0.2-0.4

Response 7: Thanks for your suggestion. Thanks for your comment. We have corrected it. This part is revised as:

Adhesiveness had a strong positive correlation with cohesiveness, springiness, fructose, glucose, soluble pectin and hemicellulose content…Springiness had strong negative correlation with starch content (R=-0.249, p<0.05).

Point 8: Line 242 “low springiness” (Group III) and line 246 “low cohesiveness” (Group IV): It seems to me that it should be “low cohesiveness” for Group III and “low springiness” for Group IV. Please check and correct if needed.

Response 8: Thanks for your suggestion. We checked and correct it. The Fig. name was wrong, then we correct it.

Point 9: Line 259: “sweet” – Your study did not evaluate the sweetness of sweet potato varieties. I suggest changing to “high soluble sugar contents”.

Response 9: Thanks for your suggestion. We accepted your suggestion, changed it as high soluble sugar contents.

In general, we think your opinion is constructive. Thanks again for your suggestion.

Round 2

Reviewer 1 Report

dear authors

thank you for applying the reviewer's comments

Author Response

Dear reviewer:

Thanks for your comments, all the authors wish you good health and good luck wish.

Best regard!

Ximing Xu
